# Community perceptions on causes of high dizygotic twinning rate in Igbo-Ora, South-west Nigeria: A qualitative study

**Akhere A. Omonkhua**[1,2☯], **Friday E. Okonofua**[2,3,4☯]*, **Lorretta F. C. Ntoimo**[5☯], **Austin I. Aruomaren**[6‡], **Ayodeji M. Adebayo**[7‡], **Roseangela Nwuba**[8‡]

1 Department of Medical Biochemistry, School of Basic Medical Sciences, University of Benin, Benin City, Nigeria, 2 Centre of Excellence in Reproductive Health Innovation (CERHI), University of Benin, Benin City, Nigeria, 3 Department of Obstetrics and Gynaecology, School of Medicine, University of Benin, Benin City, Nigeria, 4 Women's Health and Action Research Centre (WHARC), Benin City, Nigeria, 5 Faculty of Social Sciences, Department of Demography and Social Statistics, Federal University Oye-Ekiti, Oye-Ekiti, Nigeria, 6 Department of Medical Laboratory Sciences, School of Basic Medical Sciences, University of Benin, Benin City, Nigeria, 7 Ibarapa Programme, College of Medicine, University of Ibadan, Ibadan, Nigeria, 8 Department of Biological Sciences, University of Medical Sciences, Ondo, Nigeria

☯ These authors contributed equally to this work.
‡ These authors also contributed equally to this work
* feokonofua@yahoo.co.uk

**Data Availability Statement:** All relevant data are within the paper.

## Abstract

### Background

Dizygotic (DZ, non-identical) twinning rates vary widely across different regions in the world. With a DZ twinning rate of 45 per 1000 live births, Igbo-Ora Community in South-west Nigeria has the highest dizygotic (DZ) twinning rate in the world. Although several postulations exist on the causes of high DZ twinning rates in Igbo-Ora, no study has yet been conclusive on a definite causative agent.

### Objective

Using qualitative methods, this study explored the perceptions and beliefs of Igbo-Ora residents about the causes of high DZ twinning rates.

### Methods

Focus group discussion sessions and key informant interviews were organized among fathers and mothers of twins, those without twins, and health care providers. Key informant interviews were also held with persons considered to be custodians of culture who may have knowledge relevant to twinning such as traditional rulers, and traditional birth attendants; as well as health care providers, mothers and fathers of twins, and adult twins.

### Results

The results showed three factors featuring as the leading perceived causes of twinning in the community. These included twinning being an act of God, hereditary, and being due to certain foods consumed in the community. Contrary to reports that the consumption of a

**Funding:** AAO - CERHI-CMARL-002 Centre of Excellence in Reproductive Health Innovation, University of Benin, Benin City, Nigeria https://www.cerhiuniben.edu.ng/ The funders had no role in study design, data collection and analysis, decision to publish, or preparation of the manuscript. RN - TETFund/DR&D/CE/NRF/STI/36/VOL1 Tertiary Education Trust Fund (TETFund), Abuja, Nigeria https://www.tetfund.gov.ng/ The funders had no role in study design, data collection and analysis, decision to publish, or preparation of the manuscript.

**Competing interests:** The authors have declared that no competing interests exist.

species of yam (*Dioscorea rotundata*) may be responsible for the DZ twinning in this Community; yam was not prioritized by the respondents as associated with twinning. In contrast, participants repeatedly mentioned the consumption of "ilasa" a soup prepared with okra leaves (*Abelmoschus esculenta*) with water that is obtained from the community, and "amala" a local delicacy produced from cassava (*Manihot esculenta*) as the most likely dietary factors responsible for twinning in the community.

## Conclusion

Since the same foods are consumed in neighboring communities that have lower rates of twinning, we conjecture that nutritional and other environmental factors may produce epigenetic modifications that influence high DZ twinning rates in Igbo-Ora community. We conclude that more directed scientific studies based on these findings are required to further elucidate the etiology of the high rate of DZ twinning in Igbo-Ora.

## Introduction

Cultures all over the world have always been fascinated by multiple births i.e. the delivery of twins and higher-order multiples such as triplets, quadruplets, etc. There are two types of twins, monozygotic (MZ) arising from a single zygote that eventually divides into two (identical twins) and dizygotic (DZ) twins that arises from two separate zygotes (non-identical twins) [1]. The global incidence of MZ twins is relatively constant at 4 per 1000 births [2]. However, the incidence of DZ twins varies from one region of the world to another. In the 1970s, western European countries recorded twinning rates of 9–11 per 1,000 [3] while Japan, Hong Kong, and Singapore had twinning rates of 5 to 6 per 1000 births [4]. In Nigeria, twinning rates of 33–66.5 per 1000 births in Yoruba women in Western Nigeria [5,6] and 19.4 in Hausa women in the Northern part of Nigeria [5] have been reported.

Several factors are believed to contribute to DZ twinning; these include genetic predisposition [7,8], race [7], increased maternal height, increased maternal age, parity, and nutrition [9]. Also, high twinning rates in countries such as the USA, Europe, Australia, and Asia [10–13], have been attributed to the onset of medically assisted reproduction and increased maternal reproductive age [8,14].

For decades, it has been established that the Yoruba ethnic group in South-west Nigeria has the highest spontaneous dizygotic twinning rate in the world (4.4% of all births) [15]. Despite the documented initial cultural reluctance at the birth of twins in Yorubaland, there is now evidence that many Yoruba customs now revere and celebrate twins [16,17]. Initial research by Nylander in Igbo-Ora, a community in Ibarapa Central Local Government Area of Oyo state, South-west Nigeria, recorded an average of 45 to 50 sets of twins per 1000 births [18]. More recent studies have reported dizygotic twinning rate of 1 twin in 22 births [19] in this community, about four times higher than figures from anywhere else in the world. Similar high twinning rates have also been reported in other predominantly Yoruba areas such as Ekiti and Ilesha [20]. By contrast, twinning rates in the eastern [21,22] and northern parts [23,24] of Nigeria and in Ghana [25] are significantly lower.

It has long been established that higher twinning rates among the Yorubas in Nigeria are observed in women of lower social class [6,26]. This led to the hypothesis that the dietary intake of this low socioeconomic class of women may be responsible for the high twinning rates. In 1978, Nylander suggested that some substance in the diet (e.g. yams) of the Yorubas

may cause high serum concentrations of follicle stimulating hormone. An editorial published in the Lancet in 2006 [27] gave further impetus to the idea of diet being the dominant factor in the etiology of DZ twinning but to date, no substantive investigations have been undertaken to confirm or refute the hypothesis.

The studies that have been conducted to elucidate the causes of high DZ twinning rates in Western Nigeria have shown that the high incidence is not due to a difference in the maternal age and parity structure of the Nigerian population, as compared with European populations [19,28]. Indeed, when corresponding age and parity groups in the Western Nigerian and British populations were compared, the Nigerian twinning rates were still approximately four times higher [29]. There were, however, marked differences when the incidence was related to social class. In the British population there was no difference in DZ twinning rates between social classes. In contrast, the DZ twinning rate in the Nigerian population was highest in the lowest social class (approximately 62 per 1000 maternities), and lower in the highest social class at only about 15 per 1000 maternities [30]. The theory of a major environmental factor (probably dietary) as a cause of the high DZ twinning rate in Western Nigeria is further strengthened by the findings related to social class. The Nigerian women in the lowest social class (who eat mainly the local or "native" diet) had a very high twinning rate whereas those in the highest social class (who would eat far more "European diet" than most of the population) had a twinning rate similar to that in Europe [6,26,31]. In hospital based surveys, Marinho et al [32] reported a decline of twinning rates in Igbo-Ora to 23.8 per 1000 from 45–53 per 1000 maternities reported by Nylander in 1969. The authors speculated that rapid changes in dietary patterns may be responsible for the decline. Another study reported that the twinning rate of Japanese people who moved to California doubled [33]. This lays some credence to the speculation that environmental factors may play important roles in the incidence of DZ twinning.

Despite the enormous postulations on the possible causes of the high rate of twinning in Igbo-Ora, there has been no conclusive evidence pointing to a definitive causative factor. We believe that an initial exploration of the perceptions and beliefs of residents who have lived in the area for generations will throw more light to enable further and more detailed investigation of this intriguing phenomenon. It is against this background, that this qualitative study was done to ascertain the perceptions of community members regarding the causes of high DZ twinning rate in Igbo-Ora. We believe that the results of the study will provide an initial basis for more directed scientific studies for elucidating the etiology of this occurrence.

## Methods

### Study site

Nigeria is the most populous country in Africa with an estimated population of 201 million and a total fertility rate of 5.3 [34,35]. The country has a population policy that recommends four children per woman but it is hardly enforced [36]. With an annual population growth rate of 3%, Nigeria is projected to be the third most populous country in the world in 2050 [35,37]. The country has over 350 ethnic groups and distinct cultures that accord high value to childbearing and large family size [34,38,39]. Multiple births are desired and viewed as a blessing by many ethnic groups in the country. Administratively, Nigeria consists of 36 states and a capital territory, Abuja. Each state is made up of Local Government Areas. This study was conducted in Igbo-Ora, the headquarters of Ibarapa Central Local Government Area of Oyo State, South-west Nigeria, located about 80kms north-west of Lagos. Igbo-Ora is rural and most of the indigenes are peasant farmers and artisans. A *de jure* census conducted 2013 by the Department of Community Medicine, University of Ibadan, Nigeria reported a population of 64,431

people for Igbo-Ora [40,41]. With a Nigerian population growth rate of 3%, the current population estimate is 78,865 persons.

## Study design and study population

This was a community-based study that employed qualitative methods—Focus Group Discussions (FGDs) and Key Informant Interviews (KIIs). The use of these two methods was to enable the generation of a wide range of perspectives on the perceived causes of dizygotic twinning. The participants included adults aged 18 years or more. The participants for KIIs were persons who are considered custodians of culture and knowledge relevant to the experience of twinning. Thus, we recruited traditional rulers, traditional birth attendants, health providers (Chief Medical Records Officer and Senior Community Health Worker), fathers and mothers of twins, and adult twins.

The FGDs were organized among fathers of twins, and mothers who have twins and those who do not have twins, as well as health providers (one nurse, one health assistant, one senior community health worker, and five health record officers). The participants were recruited purposively by officials of Ibarapa Community and Primary Health Care Programme, College of Medicine, University of Ibadan, Nigeria who assisted in coordinating the project. These officials, some of whom are indigenes of the study community, have worked in the study community for many years and are familiar with the community and the members.

## Data collection procedure

In-depth Interviews with key informant and focus group discussion guides were developed by the research team. The interview tool included a list of open-ended questions to explore the respondents' opinion and beliefs about the factors responsible for high dizygotic twinning in Igbo-Ora. The tool was developed to start with the most factual and easy-to-answer questions, followed with those questions that solicited the informant's opinions and beliefs about the topic.

Five postgraduate students from the University of Benin, Nigeria conducted the interviews. They participated in the research meetings that developed the FGD and KII guides and were trained for two days on interview techniques. Thereafter, pilot FGDs and KIIs were conducted by the interviewers in Ado-Ekiti, Ekiti State, Nigeria and the guides were revised to accommodate the findings of the pilot study. Before the commencement of the study, six field workers were recruited and trained in Igbo-Ora to facilitate the interviews especially those conducted in Yoruba language. Subsequently, the interviews and FGDs were conducted such that one interviewer from the research team was present at every interview. Apart from note taking for each interview; all interviews were tape recorded. All the data collection/interviews took place from February 3 to February 7, 2020. Apart from the KII of traditional rulers which took place in their palaces; all other KIIs and FGDs were done in the Ibarapa Town Hall and the premises of Ibarapa Community and Primary Health Care Programme, College of Medicine, University of Ibadan in Igbo-Ora. The number of KIIs and FGDs was determined by data saturation. A total of eleven KIIs, and 7 FGD sessions, comprising 71 participants, were conducted. Each FGD session consisted of 6–12 persons [42], and lasted 60–90 minutes whereas the KIIs lasted 45–60 minutes. To achieve better group interaction, FGD for women was organized along the focus of age, and the FGD for health providers included workers in the primary health care centres in the LGA. A description of the participants and the groups is presented in Table 1.

## Data analysis

All interviews and FGDs were transcribed verbatim. The interviews in Yoruba were also transcribed verbatim and then translated to English. All transcribed interviews were checked for

**Table 1. Focus group discussions (FGDs) and key informant interviews (KIIs) conducted in Igbo-Ora.**

| Number of contacts | Study Population | Age Group | Number of Participants |
|---|---|---|---|
| | **Focus group Discussions** | | |
| 2 | Women who have given birth to dizygotic twins | 36–40 | 9 |
| | | 41 and above | 10 |
| 2 | Women who have not given birth to dizygotic twins | 36–40 | 8 |
| | | 41 and above | 8 |
| 2 | Fathers of dizygotic twins | - | |
| | FGD 1 | | 9 |
| | FGD 2 | | 8 |
| 1 | Health care providers | - | 8 |
| | **Key Informant Interviews** | | |
| 2 | Traditional ruler | - | 2 |
| 2 | Health care providers | - | 2 |
| 1 | Traditional birth attendant | - | 1 |
| 2 | Mother of dizygotic twins | - | 2 |
| 2 | Father of dizygotic twins | - | 2 |
| 2 | Mature dizygotic twins | - | 2 |
| Total number of participants | | | 71 |

correctness. Data from the interviews were analysed with the aid of Atlas.ti version 6.2.25 using the thematic analysis approach to qualitative data analysis. The transcripts were read line-by-line. Statements that are related to the cause of dizygotic twinning and related narratives were identified and coded according to themes emerging from the narratives. Related codes were merged. The findings were validated through reasoned consensus by the researchers and triangulation of methods.

## Ethical considerations

Permission to conduct the study was obtained from the Chairman, Ibarapa Central Local Government Area (IBCLGA), Igbo-Ora, Oyo State and the Director, Ibarapa Programme, College of Medicine, University of Ibadan, Ibadan, Oyo State, Nigeria. Written informed consent was obtained from all participants after the objectives of the study had been explained. All personal identifiers were removed from the transcripts. Ethical Approval for the study was provided by the Ethical Review Committee of the Faculty of Pharmacy, University of Benin (EC/FP/019/18).

## Results

### Perceived causes of high dizygotic twinning in Igbo-Ora

Several themes emerged from the interviews and focus group discussions on the perceived causes of twinning in Igbo-Ora. All the participants admitted that twinning is a common phenomenon in their community, indeed, according to one of them "there is no family in Igbo-Ora that does not have at least one twin".

> My mother is a twin; my Mom gave birth to twins; I sitting here am a twin; I also gave birth to twins; and one of my siblings also gave birth to twins. My mom's junior sister also gave birth to twins. My husband's family also has a twin birth. My father's family also has twin-

births, so there are many twins; there are many twins in Igbo-Ora. I have not seen a compound you will get to where there are no twins (KII Mature Twin I).

There is nothing else. Just like I have said before, there is no compound without twins in Igbo-ora. My mother has twins; my elder sibling has three (3) sets of twins; the next one has twins; I have twins; and the next one after me has twins. In fact, out of the six (6) children my parents gave birth to, only one (1) didn't give birth to twins. It is almost as if it is in the blood (Respondent 2, FGD2 Fathers of twins).

Although none of them could give account of how the high prevalence of twins in Igbo-Ora started, they all had stories to tell about what causes it. Some of them spoke with a high sense of certainty, others seemed speculative, with a few dissenting voices.

**Attribution to God.**   The majority of the participants attributed the high occurrence of dizygotic twins in Igbo-Ora to an act of God. Many of those who held this view believe it is a divine destiny—that God created Igbo-Ora people specially to give birth to twins. Others viewed the high prevalence of twins as a sign of God's special love and blessing to Igbo-Ora people. Attributing twinning in Igbo-Ora to an act of God was mentioned in 11 out of the 18 FGDs and KIIs.

Igbo-Ora is a town of twins; I gave birth to twin; my younger sister did; and in fact one of my sisters gave birth to twins twice. But the thing is, it is God that has blessed Igbo-Ora with twins because right now other villages around us eat the same thing we eat but it's only Igbo-Ora that gives birth to twins; so it's just a blessing from God (Respondent 3, FGD1 fathers of twins).

You see the twins that we give birth to in Igbo-Ora community is a gift from God. Anybody that God has planted the fruit of twins inside, must surely give birth to twins. Sometimes such people give birth to twins, two, three or four times. I and all my younger sisters have given birth to twins and also my mother (Respondent 5, FGD mothers with twins 36–40 years old).

**Attribution to genetics/hereditary.**   Closely related to the perception that twinning in Igbo-Ora is an act of God, some of the participants attributed it to genetics and hereditary. Attributing the high rate of twinning to hereditary, many of them alluded to what they called "bond or cord of twins" in Igbo-Ora people. They claimed that only people who have the cord of twins give birth to twins. Some of them revealed that it is not a common practice for Igbo-Ora people to inter-marry with other persons of Yoruba origin and ethnic groups. This suggests that the high rate twinning in Igbo-Ora is sustained by their tradition of endogamy. When two persons who have the "bond of twins" marry, the result will be more twins. Some of the participants claimed that they give birth to many twins because the forefathers of Igbo-Ora were twins.

'Okun Ibeji' (bond of twins). The way I am now, since I gave birth to twins, my child too can give birth to twins. And then, the number of twins will increase. (Respondent 7, FGD mothers of twins 36–40 years old).

The forefathers of Igbo-Ora were twins and that is why twins are many here.

(Respondent 6, FGD, mothers of twins 36–40 years old).

I think there is no way we can do away with hereditary. Anywhere that you have twins and people in that area marry each other, you are already tilting towards hereditary. Because if I

am a twin and I now marry someone from my town who is also a twin, of course the chances are very high or they become higher that we are going to have twins. Mind you, most of them (Igbo-Ora indigenes) marry each other. They marry among themselves. It's not as if they do not marry outsiders but 90% of them marry each other. So, that is a very strong factor... (KII Health care provider I).

**Attribution to dietary intake.   Okra leaves (*Ilasa*) and yam or cassava flour (*Amala*).** Others attributed high twinning to a common dish in Igbo-Ora—soup made with *ilasa* (okra leaf) and eaten with *amala* (yam or cassava flour). This perception was mentioned in 13 out of the 18 FGDs and KIIs in 26 quotations (Fig 1 and Table 2). The narratives of the participants who hold this view indicated that *ilasa* has to be eaten with *amala* to produce the twinning effect.

I greet everyone today (prayers in Yoruba). Twinning is a very important thing in Igbo-Ora, because all around Ibarapa, Igbo-Ora is still number one when it comes to twins. The reason for this is because we use *ilasa* to eat *amala* a lot, and twins do come each time we do that (Respondent 1, FGD1 fathers of twins).

We take *ilasa* together with cassava flour, while other people add other leaves to *ilasa*. We eat direct *ilasa* here, and we have two (2) types. We have *ilasa* "Iloko" and ilasa "dokoso"." (KII Healthcare provider).

**Ilasa.**   There was a category of respondents who specifically reported that the cause of twinning is *ilasa*. This notion was mentioned in 15 out of the 18 contacts (Table 2). To them, *ilasa* is a cause of twinning irrespective of what it is eaten with. However, they emphasized that it is the particular species that is found in Igbo-Ora that produces the twin birth effect.

I know that you people are wondering why there are so many twins in Igbo-Ora, wondering what the cause is. It is caused by the food we eat (Respondent 5, FGD mothers of twins 36–40 years old).

Another respondent in the group interjected with the answer:

It is the *ilasa*. If your wife knows how to cook it, and make love, you will have twins (Respondent 4, FGD mothers of twins, 36–40 years old).

When our forefathers came to Igbo-Ora, one of the things they brought with them was *ilasa* leaves. Because in the whole of Nigeria, if you ask who cook *ilasa* the most, you will discover that it is Igbo-ora. Igbo-ora cooks *ilasa* the most. Then God used the love of *ilasa* to translate to the love of twins (Respondent 4, FGD mothers of twins 41+ years old).

We have *Ila* (Okra), and *Ilasa* (Okra leaf). The different types of *ilasa* (Okra leaf) that we have are *ilasa* iloko, is the one that has green color." (FGD 1 Fathers of twin)

Those who believe *ilasa* is the cause of twinning also emphasized that the Igbo-Ora *ilasa* must be cooked with water from Igbo-Ora to produce the desired effect. They claimed that when *ilasa* from Igbo-Ora is taken to other locations and cooked with water from other places, the *ilasa* does not turn out as good as it does with Igbo-Ora water.

The water here is different. The *ilasa* that we cook here, that is slimy and viscous, isn't the same in other places. It is only when they use a keg to take water from here to those other places that it turns out well (Respondent 8, FGD 1 Father of twins).

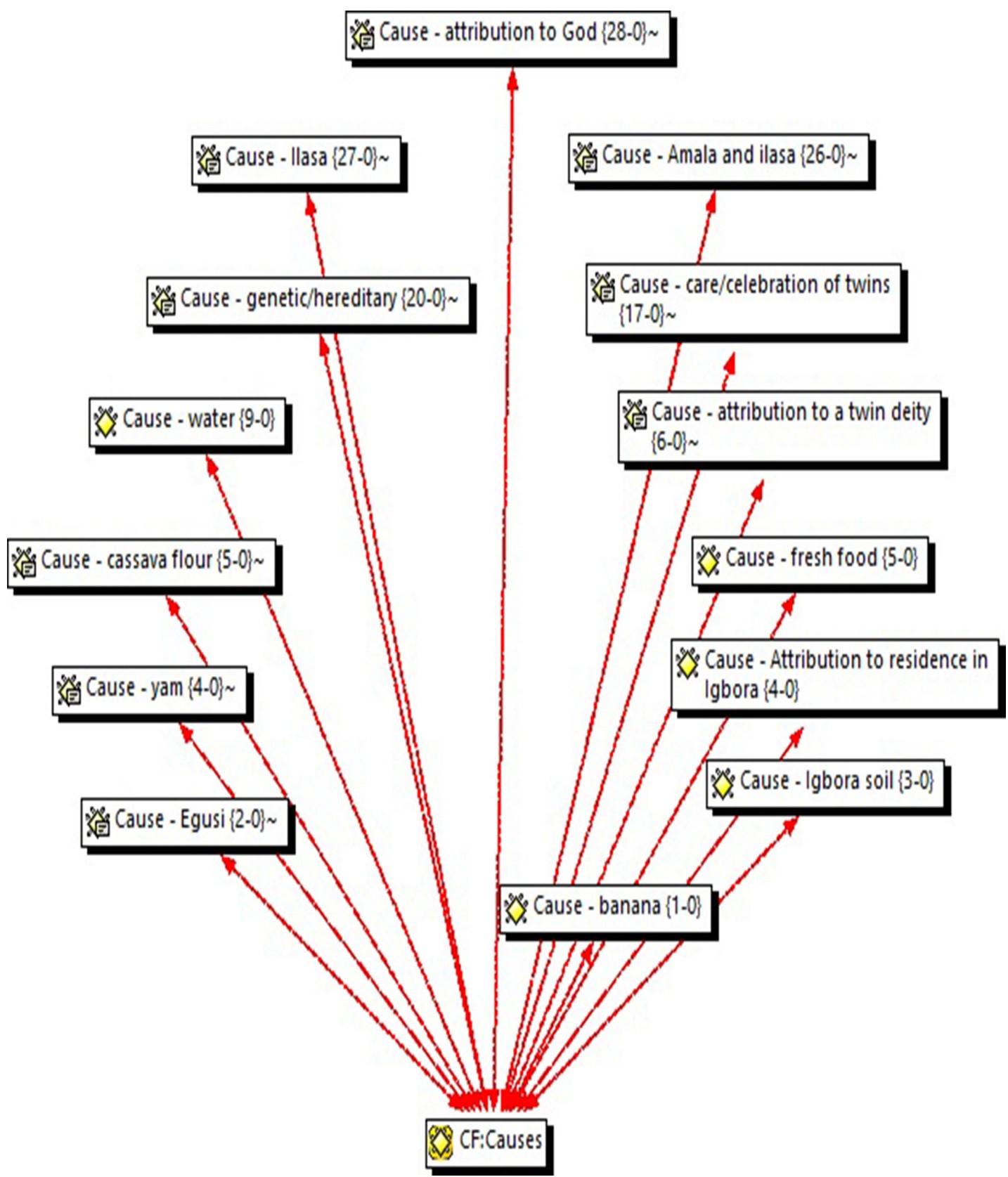

**Fig 1. Perceived causes of dizygotic twinning in Igbo-Ora with the number of quotations.**

**Table 2. Frequency distribution of theme by type of contact.**

| Type of contact | Amala & ilasa | Twin deity | God | Residence in Igbo- Ora | Banana | Care & celebration of twins | Cassava flour | Egos | Fresh food | Gene-tics | Soil | Ilasa | Water | Yam |
|---|---|---|---|---|---|---|---|---|---|---|---|---|---|---|
| FGD1- Fathers of Twins | + | 0 | + | 0 | 0 | + | 0 | 0 | + | 0 | 0 | 0 | + | + |
| FGD2—Fathers of Twins | + | 0 | + | + | 0 | + | 0 | 0 | + | + | 0 | + | + | + |
| FGD—Healthcare Provider | + | 0 | 0 | + | 0 | + | + | 0 | 0 | + | 0 | + | + | 0 |
| FGD—Mothers of Twins (36–40) | + | 0 | + | 0 | 0 | 0 | 0 | + | 0 | + | 0 | + | 0 | 0 |
| FGD—Mothers of Twins (41 and above) | + | + | + | 0 | + | + | + | 0 | 0 | + | 0 | + | 0 | 0 |
| FGD—Mothers Without Twins (36–40) | + | + | + | 0 | 0 | 0 | 0 | 0 | 0 | + | 0 | + | 0 | 0 |
| FGD- Mothers Without Twins (41 and above) | + | 0 | + | 0 | 0 | 0 | 0 | 0 | 0 | 0 | 0 | + | 0 | 0 |
| KII—Father of twins I | 0 | + | + | 0 | 0 | 0 | 0 | + | 0 | 0 | 0 | + | 0 | 0 |
| KII—Father of twins II | + | + | + | 0 | 0 | 0 | 0 | 0 | 0 | 0 | 0 | + | 0 | 0 |
| KII—Mature Twins I | 0 | 0 | + | 0 | 0 | 0 | 0 | 0 | 0 | + | 0 | + | 0 | 0 |
| KII—Mature Twins II | 0 | 0 | + | 0 | 0 | 0 | 0 | 0 | 0 | 0 | 0 | + | 0 | + |
| KII—Mother of twins I | + | 0 | + | 0 | 0 | 0 | 0 | 0 | 0 | + | 0 | 0 | 0 | 0 |
| KII—Mother of twins II | + | 0 | 0 | 0 | 0 | 0 | 0 | 0 | 0 | + | 0 | + | 0 | 0 |
| KII—Traditional Birth Attendant 1 | + | 0 | 0 | 0 | 0 | + | 0 | 0 | 0 | + | 0 | + | 0 | 0 |
| KII—Traditional Ruler I | 0 | + | 0 | 0 | 0 | 0 | + | 0 | 0 | 0 | 0 | + | 0 | 0 |
| KII—Traditional Ruler II | + | 0 | 0 | 0 | 0 | 0 | 0 | 0 | 0 | + | 0 | + | 0 | + |
| KII -Health Care Provider I | 0 | 0 | 0 | + | 0 | 0 | 0 | 0 | 0 | + | + | + | + | 0 |
| KII -Health Care Providers II | + | 0 | 0 | 0 | 0 | 0 | 0 | 0 | 0 | + | + | 0 | + | 0 |
| Total mention of theme by PD | 13 | 5 | 11 | 3 | 1 | 5 | 3 | 2 | 2 | 12 | 2 | 15 | 5 | 4 |

Note: + means theme mentioned; 0 means theme not mentioned.

You see, about the *ilasa* soup matter, whenever we go to Lagos to celebrate with them, we carry the *ilasa* soup things and keg of water, because the water in Lagos is not like Igbo-Ora's; it can't cook *ilasa* well (Respondent 7, FGD mothers without twins 36–40 years old).

However, there were a few dissenting voices of those who regarded the view about *ilasa* as a myth.

It just pleases God to allow many twins in Igbo-Ora. Just that people do say if you don't eat *ilasa*, you can't give birth to twins, I see that as mere fallacy because not everyone who eats *ilasa* gives birth to twins. In my own view, not all who eat *ilasa* gives birth to twins (KII Mature twin II).

**Igbo-Ora water.**   Some of the respondents expressed the conviction that twinning in Igbo-Ora results from the water in the community. In addition to using it to cook, it must be used for other purposes for the desired effect. In the FGD with healthcare providers, many respondents alluded to the water in Igbo-Ora as the cause of twinning.

In my own experience, I would like to differ. I want to believe that probably it's the water. Because if you take these things away from here to other places, nothing will happen and all these things we have just mentioned there is none of them you can eat raw, most of them have to be cooked and there is nothing you cook without water. So if you tell me that it's *ilasa* (Okra leaf) that is causing it . . . if I pick it from here and continue to eat, I think it should replicate what is happening here, but that has not been the case. Some people who desire twins come here and take it home every weekend and cook with it. They have done so repeatedly for many years, yet they didn't have twins. But immediately they step into this town and come and reside here and eat ilasa, they begin to have twins. That residency aspect should be emphasized. Once you reside here, and it's not as if immediately you reside here you automatically have twins, but majority of those who happen to reside here get twins. (KII, Healthcare provider I).

**Residence in Igbo-Ora.**   The above view by the health provider suggests that physical residence in Igbo-Ora may be a contributory factor. The view is that if Igbo-Ora *ilasa* is the cause of twinning, then it should work anywhere you take it but it does not seem to be the case. They think residence as well as eating the common food *ilasa* and using their water increases the chance of twin birth.

If you go to Lagos, you may have twins if it happens that you go there (Igbo-Ora) one or two years, if after that, your major place is Lagos now and you are there for years again, you may not get twins because you have already abandoned what you have been taking from your home town. But if you are here, you stay here for 2 or 3 years even if you are not an indigene, you will have twins definitely. I worked with one Oyinbo [White woman] who stayed here for several years and who eventually delivered two twins (Respondent 3, FGD health providers).

A moderator asked: "When non-natives come to Igbo-Ora and live with you, do they give birth to twins? Are there examples?" The response in the FGD for fathers of twins was a chorus yes. Some of the respondents gave examples.

When they eat what we eat and drink our water, they will also have twins (Respondent 3, FGD 2 fathers of twins).

Someone from Agatu (Benue State) came to Igbo-Ora. He lived in my house. Then his wife got pregnant. When she eventually delivered, she gave birth to two (2) boys (Respondent 5, FGD 2 fathers of twins).

**Care and celebration of twins.**   Twins are celebrated in Igbo-Ora. The twin day celebration in Igbo-Ora attracts people from within and outside Nigeria. Apart from the twin day celebration, Igbo-Ora indigenes eulogize twins. Some of the respondents think twin births are common in Igbo-Ora because the people celebrate and accord special care to twins. In fact, the care of twins is in the form of deification for some Igbo-Ora indigenes. Twins are seen as special persons that should be treated with care and deified. Beans pottage is cooked when twins are born and used to entertain twins. Some of the respondents claim that the beans pottage

invites twins to them. Others submitted that showing kindness to twins and giving them and giving gifts to their parents attract more twinning.

> I think it is the way we take care of twins. People celebrate twins on their own, privately, before they bring them out for the general celebration. When it is time for the celebration, people cook beans for the twins (KII, Traditional birth Attendant).

> In Igbo-ora that we are, when we are celebrating twins. . . they used to cook beans pottage. This made twins abundant in Igbo-ora. We use it to give commands for them to come to us (Respondent 7, FGD1 Fathers of twins).

A respondent narrated a personal experience

> When you take care of twins, they become abundant in your household. I am saying this because my father's child has twins. I used to beat them because they were stubborn. I wasn't pregnant then. I really used to beat them, then one day, they came to my house. I started wondering about what to give them. I treated them well on that day; I asked if they had eaten, they replied in the negative; and it was 6:50 pm or 7pm. So, I went to buy beans, I bought it and they asked for garri, which I gave them. They also asked for palm oil, I was going to use the palm oil to cook, but I decided to add it to their food. I asked if it was enough, they affirmed. Then they finished eating, they stood to take their leave. They were about six years old then; they thanked me, then told me to expect them in my house. They said they would be many in my house. I asked "many?" they said yes, that they will visit my household. Eventually, I got pregnant and delivered twins. They then came to my house to remind me that they said they would visit my household (Respondent 9, FGD mothers of twins 41+ years old).

There are special songs for twins in Igbo-Ora. They believe that praising twins attracts more twinning.

> The first is the mercies of God. The second one is that whenever they see twins from afar, they start praising them. These praises will get to the heads of the twins and within them, they'll say 'this clan should get a twin or more twins. Since they like them so much'. Even yesterday, someone is even expecting it. Twins are children that elicit favours. That's why they are increasing in Igbo-ora (Respondent 4, FGD mothers of twins 41+ years old).

When asked about the song for twins, one of the mature twins sang:

> There is palm oil, there is beans. I am not afraid.

> I am not afraid to give birth to twins 2ce.

> There is palm oil, there is beans.

> (KII, Adult Twin).

**Attribution to a deity (god of twins).** Closely related to celebration of twins is the worship of the gods of twins. Some respondents attributed the high prevalence of twins in Igbo-Ora to a deity whose worship assures the survival of twins and sustains the high prevalence.

> All that everyone has said so far is the truth. However, in the olden days, temples/shrines were built for twins. As I am seated here the shrine for worshipping twins which I inherited

from my mother is what I am still using, which is why I gave birth to twins. When our mothers cook beans, they also carry out certain rites (which she goes on to list). When it is about three (3) months, I cook beans, to celebrate the twins that God gave me. I speak words and I dance. I call drummers who dance well also, because they are from my god. . . (Respondent 3, FGD mothers of twins 41+ years old).

**Cassava flour.** Some of the respondents mentioned cassava flour as contributing to the high twinning in their community. They described the specific type of cassava and how it is prepared.

. . . And we have a specific kind of *amala* I mean cassava flour before you produce *amala* i.e. "odongbo". We call that cassava "odongbo"; another one is *Orkori*; while another one is *Texaco* and *Oko-iyawo*. To prepare cassava to become cassava flour, some people will peel, while others will just cut it and soak it in water, so after 3 days they air-dry it. They then dry it in a cassava drying platform, before grinding it. So mostly that is what our people depend on; we don't usually eat "*amala-asun*" (the broom amala); we eat "elubor"; we also call it "pellumi". The reason why they call it *pellumi* is because when you take it, you cut it like this (demonstrating) and drop it inside the water. You don't peel it, so it's eaten with the bark (Respondent 3, FGD Health providers).

**Fresh food.** The narratives of some respondents indicate that their farming occupation predisposes them to twinning because they plant what they eat and eat them fresh. Respondents who hold this view submitted that eating *ilasa* and *amala*, and other types of food fresh is a reason for high twinning in Igbo-Ora.

The food and work we do is still one of the reasons, we are farmers, everything we eat is all fresh, it's not like it has been canned, even the *ilasa* we get it from the farm, the yam powder we make it ourselves, a lot of things are the reasons why twins are many here (Respondent 2, FGD1 Fathers of twins).

**Yam.** Those who mentioned yam (*Dioscorea rotundata*) was in the context of using it to prepare beans pottage and to make *amala* (yam flour). One traditional ruler stated that a type of yam may be a reason for twinning.

Some nutritional people from Ibadan actually worked on that and it was confirmed that there is one *Agida*—yam that you know. We have 2 or 3 types of *amala*. We have *amala* made from cassava, and some from yam. Most people don't prepare *amala* from yam because it is a bit sophisticated and a bit costlier than the *amala* from cassava. So, it was claimed in some experiments that have been done that there are some enzymes inside yams that can produce multiple eggs and this one can assist multiple births (KII, Traditional ruler II).

**Igbo-Ora soil.** Some of the respondents were of the view that the soil on which *ilasa*, and the other types of food that are suspected to cause twinning in Igbo-Ora, is planted may be the actual reason. They think that the soil in Igbo-Ora plays a role.

For the purpose of this interview, Igbo-ora yam, water yam, cassava and Okra might be different. I would not want to say it's also a special yam or special leaf but because they are planted here we might need to consider the possibility of soil analysis. This is because yam

is been planted in Ibadan, Benin and other places, yet they don't seem to produce twins like it is in Igbo-ora. So, we also talk about water from the soil, at least from the borehole. That still takes us back to the water again because water goes into the planting of these things. If we have to check again to look at what exactly could be responsible aside from the products itself, we might need to look at the planting with water and the soil. . . (KII, Health provider I).

**Melon.**   Melon called "egusi" in the local language was mentioned as a cause. In particular, melon leaf prepared with locust beans was viewed as a cause of twinning.

That [referring to *amala* and *ilasa*] is the cause. White *amala* and *ilasa* and *egusi* (Respondent 3, FGD mothers of twins 36–40 years old).

I do hear people talk about melon leaf prepared with locust beans (KII, Father of twins I).
**Banana.**   Banana was also mentioned as a possible cause of twinning in Igbo-Ora.

Bananas, oranges, maize, beans. They (twins) like all these things but bananas are number 1. Anywhere these things are, twins will love to be there (Respondent 5, FGD mothers of twins 41+ years old).

## Discussion

The study was designed to explore community perceptions related to the causes of DZ twinning in Igbo-Ora, South-west Nigeria. Consistent with our expectations, community members were very eager to speak about the topic and gave several explanations which they perceived as reasons for the high DZ twinning in the community. The answers they gave covered three domains–an act of God, hereditary and nutritional factors. The interpretation that God was responsible is expected, and can be linked to hereditary, especially because no specific evidence was provided to justify the reasons for the divine explanation. Thus, hereditary causes were the top-notch explanation provided by the community members for the high rate of twinning in Igbo-Ora. This was supported with statements such as "our neighboring communities do not have the same twinning rate despite that they eat the same type of food"; "women who have given birth to twins are (those) more likely to give birth to more twins"; and "our people hardly intermarry, which makes the twinning hereditary to remain in the community".

Although hereditary and genetics are factors that have been surmised as causes of high DZ twinning rates, the specific genetic markers that predispose to DZ twinning have only been partially identified in some Western countries [43,44]; such studies have not been done in Africa. It is also not clear why such genetic factors should feature more in specific communities rather than being widespread universally. It is evident therefore that more research is required to identify the specific genes or the factors that modulate them, thus predisposing to high rate of DZ twinning in the community.

In contrast, the community members spoke more specifically and in greater detail about possible nutritional factors that may explain the high rate of DZ twinning. While yams (*Dioscorea* species) have been proposed by several previous researchers as responsible for the high twinning rate in the community [18,19,27,45,46], the results from interviews and focus group discussions in this study did not rank the consumption of yams very highly. This low rating of yams appears to answer a major dilemma that had previously existed in attempts to link yams to the high DZ twinning rate which is that while yams are eaten in most parts of Nigeria, the twinning rates do not appear to be as high in other regions of the country [21–24]. This low rating of yam in this study will therefore suggest that a lower preference should be given to

further research that focuses on exploring yams as a potential causative factor in the high twinning rate in Igbo-Ora.

In this study, a type of native soup called "ilasa" prepared with okra (*Abelmoschus esculentus*) leaves featured prominently and repeatedly in responses of the participants as a major dietary cause of twinning. However, there was widespread agreement that *ilasa* has to be prepared specifically with the water obtained from Igbo-Ora; otherwise its effect on twinning would be minimal. Other food items mentioned included "amala" produced from cassava (*Manihot esculenta*), fresh foods, melon and banana with less degrees of certainty.

As evidence that there were still some unknown factors, the participants emphasized the point that residency for a significant length of time in the community will boost the chance of a twin pregnancy. Several examples were given of visitors from other parts of the country who experienced twin pregnancy only after they stayed in the community for a long period of time. Some respondents mentioned the way twins are respected and revered in the community as well as possibly other prevailing unknown factors which may increase the likelihood of twinning. The mention of prolonged residency suggests that whatever factor may be responsible, it is possibly modulated through active participation in the events in the community which may include the consistent consumption of the responsible diet in the community.

Based on the perceptions and interpretation given by community members, it is possible that the high DZ twinning rate in Igbo-Ora may be explained in epigenetic terms, which is described as available heritable phenotype changes that do not involve alterations in the DNA sequence [44,47]. It is possible that the form of diet eaten in the community may after a significant period of time, alter the expression of the inherited genetic phenotype and increase the likelihood of twinning. Only those who have lived in the community for a significant period of time will therefore have the benefits of this alteration and therefore experience the higher likelihood of twinning.

We therefore strongly believe that any further study designed to investigate the high rate of twinning in Igbo-Ora should focus on genetic and epigenetic components, and also on identifying dietary determinants. The major diets should include "ilasa", the specific water used in the community, "amala" produced from cassava products, and fresh foods eaten in the community. We also believe that an immediate case reference study comparing the dietary history and characteristics of mothers with twins with those without twins, and including the investigation of the length of residence in the community will help the further exploration of this important research question.

Although the FGDs provided a wide range of perceptions on the causes of DZ twinning in the study community, the views of the FGD participants may have been influenced by views already expressed by others in the group. However, the in-depth interviews with the same category of respondents provided insights that were used to check the content of the FGDs. Also, the narratives represent the subjective views of the participants, thus, the results of this study cannot be generalized to the entire population, and cannot be taken for the true causes of high DZ twinning in Igbo-Ora.

The major strength of this study is its use of a qualitative method of inquiry whereby community stakeholders had the opportunity to speak on the subject matter using their own words and without being prodded. From the study, it was evident that many constituents were familiar with the subject matter and were eager to speak about it. To the best of our knowledge, this is the first study of a qualitative nature that explores reasons for the high rate of twinning in Igbo-Ora in Southwest Nigeria. The results, although not substantive, they nevertheless provide important leads for interested researchers, and have implications for the design of further epidemiological and experimental studies that explain the reasons for the highest rate of twinning ever reported anywhere around the world.

## Acknowledgments

The authors acknowledge the immense support received from Mr. Ayodeji Oduniake and staff of Ibarapa Community and Primary Health Care Programme. The authors also acknowledge all postgraduate students (Kevin A Akonfua, Aisosa Eguavoen, Joseph Omo-Erhabor, and Brian Igboin) and field assistants who participated in data collection. The authors are particularly grateful for the overwhelming support received from the Igbo-Ora community leaders as well as all those who participated in the study.

## Author Contributions

**Conceptualization:** Akhere A. Omonkhua, Friday E. Okonofua, Roseangela Nwuba.

**Data curation:** Akhere A. Omonkhua, Friday E. Okonofua, Lorretta F. C. Ntoimo, Austin I. Aruomaren, Ayodeji M. Adebayo, Roseangela Nwuba.

**Formal analysis:** Friday E. Okonofua, Lorretta F. C. Ntoimo.

**Funding acquisition:** Akhere A. Omonkhua, Roseangela Nwuba.

**Investigation:** Akhere A. Omonkhua, Friday E. Okonofua, Lorretta F. C. Ntoimo, Austin I. Aruomaren, Ayodeji M. Adebayo, Roseangela Nwuba.

**Methodology:** Akhere A. Omonkhua, Friday E. Okonofua, Lorretta F. C. Ntoimo, Austin I. Aruomaren.

**Project administration:** Akhere A. Omonkhua, Friday E. Okonofua, Ayodeji M. Adebayo, Roseangela Nwuba.

**Resources:** Akhere A. Omonkhua, Friday E. Okonofua, Austin I. Aruomaren, Ayodeji M. Adebayo, Roseangela Nwuba.

**Software:** Lorretta F. C. Ntoimo.

**Supervision:** Akhere A. Omonkhua, Friday E. Okonofua, Lorretta F. C. Ntoimo, Ayodeji M. Adebayo, Roseangela Nwuba.

**Validation:** Akhere A. Omonkhua, Friday E. Okonofua, Lorretta F. C. Ntoimo, Austin I. Aruomaren.

**Visualization:** Lorretta F. C. Ntoimo.

**Writing – original draft:** Akhere A. Omonkhua, Friday E. Okonofua, Lorretta F. C. Ntoimo.

**Writing – review & editing:** Akhere A. Omonkhua, Friday E. Okonofua, Lorretta F. C. Ntoimo, Austin I. Aruomaren, Ayodeji M. Adebayo, Roseangela Nwuba.

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
