## [Decision Letter · Decision Letter 0]

26 Oct 2020

PONE-D-20-17981

Community Perceptions on Causes of High Dizygotic Twinning Rate in Igbo-Ora, South-west Nigeria: A qualitative Study

PLOS ONE

Dear Dr. Okonofua,

Thank you for submitting your manuscript to PLOS ONE. After careful consideration, we feel that it has merit but does not fully meet PLOS ONE’s publication criteria as it currently stands. Therefore, we invite you to submit a revised version of the manuscript that addresses the points raised during the review process.

We look forward to receiving your revised manuscript.

Kind regards,

Juliet Kiguli, MA, PhD

Academic Editor

PLOS ONE

Journal Requirements:

Additional Editor Comments:

Please address the reviewers' comments .

Reviewers' comments:

Reviewer's Responses to Questions

**Comments to the Author**

1. Is the manuscript technically sound, and do the data support the conclusions?

Reviewer #1: Yes

2. Has the statistical analysis been performed appropriately and rigorously? 

Reviewer #1: N/A

3. Have the authors made all data underlying the findings in their manuscript fully available?

Reviewer #1: Yes

4. Is the manuscript presented in an intelligible fashion and written in standard English?

Reviewer #1: Yes

5. Review Comments to the Author

Reviewer #1: METHODS: Under study design, the authors should state clearly the categories of health workers that participated in the study(page 9 line 187). Under data collection, page 9,lines 199-203the authors should state where the interviews were held and where.

RESULTS: How many people in total participated in the study with their designations for clarity.

DISCUSSION: Did the study have any limitations?

6. PLOS authors have the option to publish the peer review history of their article (what does this mean?). If published, this will include your full peer review and any attached files.

Reviewer #1: No

---

## [Author Response · Author response to Decision Letter 0]

7 Nov 2020

The following have been revised:

The manuscript has been revised according to journal style

2. Reviewer #1: METHODS: Under study design, the authors should state clearly the categories of health workers that participated in the study (page 9 line 187). 

These have been specified on page 9, lines 181 to 185 (Unmarked revised manuscript)

3. Under data collection, page 9, lines 199-203 the authors should state where the interviews were held and where.

This is clarified on page 10; lines 206 to 209

4. RESULTS: How many people in total participated in the study with their designations for clarity.

A revised table has been inserted on page 11. The total number of participants is clarified on the table and on page 10; line 211

5. DISCUSSION: Did the study have any limitations?

A section on limitations has been included. Page 33; lines 647 to 653

---

## [Editor Report · Decision Letter 1]

17 Nov 2020

Community perceptions on causes of high dizygotic twinning rate in Igbo-Ora, South-west Nigeria: A qualitative study

PONE-D-20-17981R1

Dear Dr. Okonofua,

We’re pleased to inform you that your manuscript has been judged scientifically suitable for publication and will be formally accepted for publication once it meets all outstanding technical requirements.

Kind regards,

Juliet Kiguli, MA, PhD

Academic Editor

PLOS ONE

Additional Editor Comments (optional):

Thank you for addressing the reviewer's comments.
---

## [Editor Report · Acceptance letter]

23 Nov 2020

PONE-D-20-17981R1 

Community perceptions on causes of high dizygotic twinning rate in Igbo-Ora, South-west Nigeria: A qualitative study 

Dear Dr. Okonofua:

I'm pleased to inform you that your manuscript has been deemed suitable for publication in PLOS ONE. Congratulations! Your manuscript is now with our production department. 

Kind regards, 

on behalf of

Dr. Juliet Kiguli 

Academic Editor

PLOS ONE